# Stereodivergent 1,3-difunctionalization of alkenes by charge relocation

Bogdan R. Brutiu[1,3], Giulia Iannelli[1,3], Margaux Riomet[1], Daniel Kaiser[1] & Nuno Maulide[1,2 ✉]

Alkenes are indispensable feedstocks in chemistry. Functionalization at both carbons of the alkene—1,2-difunctionalization—is part of chemistry curricula worldwide[1]. Although difunctionalization at distal positions has been reported[2–4], it typically relies on designer substrates featuring directing groups and/or stabilizing features, all of which determine the ultimate site of bond formation[5–7]. Here we introduce a method for the direct 1,3-difunctionalization of alkenes, based on a concept termed 'charge relocation', which enables stereodivergent access to 1,3-difunctionalized products of either *syn-* or *anti-*configuration from unactivated alkenes, without the need for directing groups or stabilizing features. The usefulness of the approach is demonstrated in the synthesis of the pulmonary toxin 4-ipomeanol and its derivatives.

Alkene functionalization reactions are a staple of every undergraduate programme in chemistry and can be broadly divided into two families: (1) transformations that result in either temporary (for example, olefin metathesis) or permanent (for example, ozonolysis) C=C bond cleavage and (2) reactions resulting in products that maintain the original C–C $\sigma$-bond connectivity[1,8]. Among the latter, the broad family of 1,2-difunctionalization reactions[9,10] ranges from classical halogenation to sophisticated transition-metal-catalysed processes[11–13], and the nascent field of remote functionalization has emerged as a way of redirecting reactivity towards a distal position, away from the initial reactive site[2–4,14–20]. Indeed, a number of methods for alkene 1,3-difunctionalization using transition-metal catalysis have emerged in the literature. Such methods (Fig. 1a) invariably rely on a mechanistic handle to achieve regioselectivity: a directing group or 'stopper' (such as an arene ring that provides a benzylic resting point for the catalyst) must be embedded in the substrate, primarily because of the pronounced tendency of C($sp^3$)–M species to engage in β-hydride (β-H) elimination, resulting in the generation of Heck products[5–7,21,22]. Similarly, an extensive body of research on Friedel–Crafts-type reactions of alkenes is known (Fig. 1b)[23]. Primarily, reactions of acylium ions have been shown to afford the products of 1,2-difunctionalization, with product distribution often highly dependent on the nature of substrate, reagent and solvent (Fig. 1b, top). Intriguingly, Friedel–Crafts-type reactions of alkenes have also been found to enable remote functionalization of alkenes (Fig. 1b, bottom); such transformations, however, invariably suffer from undefined selectivity—often giving mixtures of 1,2- and 1,3-functionalization products[24–27]—or are possible only under substrate control or on specialized substrates[28–30]. Thus, although the corresponding reactivity, at its core that of the Friedel–Crafts reaction, has been explored, little is known about the factors governing the selectivity or predictability of such transformations, and the direct and general 1,3-difunctionalization of unactivated and unfunctionalized alkenes remains an unmet challenge.

We present such a difunctionalization of unactivated alkenes under simple conditions, through which olefins are converted into products in which functionalization has occurred with generality at positions 1 and 3 (Fig. 1c). We achieved this by adapting electrophilic addition to alkenes as the basis for a general, predictable and highly selective strategy termed 'charge relocation'—a synthetic logic in which incipient or localized charge at a given atom relocates to a defined position through a series of hydride shifts.

Well aware that classical electrophilic additions to double bonds hinge on immediate interception of an (incipient) positive charge by a nucleophile[23], we became intrigued by the mode of electrophilic addition to a double bond in the absence of a suitable nucleophilic species. Over the course of these investigations we eventually found that the treatment of cyclohexene with an acylium cation carrying non-nucleophilic hexafluoroantimonate as the counter anion, generated in situ from acyl chloride and silver hexafluoroantimonate, swiftly and selectively formed a product of 1,3-hydroxyacylation following hydrolytic work-up (Fig. 2a; see Supplementary Information for additional conditions surveyed during optimization of reaction conditions). Notably, the success of the reaction was found to rely heavily on the nature of the halophilic reagent, with only silver hexafluoroantimonate providing a high yield of **19** whereas other silver salts or Lewis acids gave low levels of conversion to the desired product. Most interestingly, this reaction not only resulted in 1,3-difunctionalization but it did so with exclusive selectivity for the *syn*-products over a range of different unactivated alkenes, as depicted in Fig. 2.

Following this finding we turned to investigate the scope of this transformation (Fig. 2b), initially using a phenyl-substituted acylium ion. Our survey of linear alkenes, inherently lacking the ability to form diastereomeric products, showed high yields for a range of chain lengths (**1** and **2**). Appended functional groups were also found to be tolerated, with substrates bearing a trifluoroacetate (**3**) or phthalimide (**4**) providing the desired products at good yield. Importantly, 4-phenyl-1-butene—a substrate bearing potential bias due to the presence of a benzylic position—afforded exclusively the product 1,3-difunctionalization (**5**) with the benzylic site remaining untouched.

As highlighted above, the reaction with cyclohexene was found to deliver exclusively the product of *syn*-hydroxyacylation (**6**), a fact proven to be true for a large variety of arene-substituted acylium ions

[1]Institute of Organic Chemistry, University of Vienna, Vienna, Austria. [2]Research Platform NeGeMac, Vienna, Austria. [3]These authors contributed equally: Bogdan R. Brutiu, Giulia Iannelli. ✉e-mail: nuno.maulide@univie.ac.at

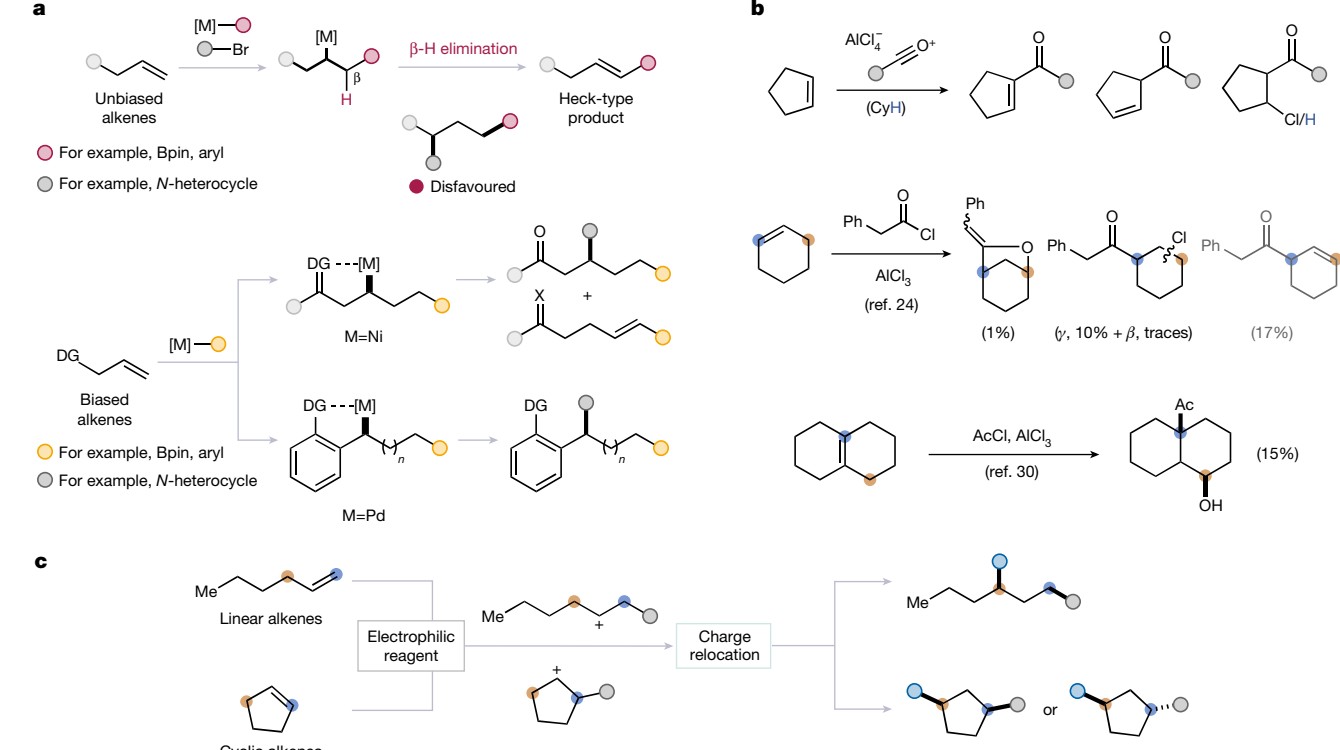

**Fig. 1 | Examples of state-of-the-art remote functionalization reactions of alkenes. a**, Current paradigm for the distal functionalization of alkenes, relying on directing-group-mediated metal catalysis. **b**, Friedel–Crafts-type reactions of alkenes generally provide mixtures of products—which depend on the substrate, reagent and solvent—with selectivity observed only with specialized substrates. **c**, This work: stereodivergent, reagent-controlled 1,3-difunctionalization of alkenes through charge relocation. DG, directing group; Bpin, pinacol boronate.

regardless of electronic or steric properties (**7**–**17**). The same stereochemical outcome was observed for a cyclopentene-derived ketoalcohol (**18**), as well as for products resulting from the addition of aliphatic acylium ions (**19**–**24**). Notably, an acryloyl-derived acylium ion also provided the desired product of *syn*-hydroxyacylation (**25**), as did those bearing heteroaromatic groups (**26** and **27**). Finally, when norbornene was employed, the intermediately formed, positively charged species underwent rearrangement (presumably via a non-classical carbocation), affording crystalline **28**.

Figure 2c shows a comparison of this method with previously reported approaches to alkene 1,3-difunctionalization, relying on substrate rather than reagent control. Indeed, when a fully unbiased alkene (1-nonene) was subjected to established, transition-metal-catalysed protocols using either nickel or palladium catalysis, no products of 1,3-difunctionalization were obtained—rather, products of Heck coupling or non-specific decomposition were observed (see Supplementary Information for details on these reactions). By contrast, the protocol presented herein—as shown above—provided the product of 1,3-difunctionalization (**2**) at 64% isolated yield.

In subsequent investigations we found that, if the final work-up was preceded by treatment with dimethyl sulfoxide (DMSO)[31], the corresponding products of *anti*-hydroxyacylation were obtained with excellent stereoselectivity (**29**–**37**, typically over 20:1 diastereomeric ratio (d.r.); Fig. 3), thus enabling flexible access to either isomer in a stereodivergent manner[32–37]. Replacement of the hydrolytic work-up by the addition of other nucleophilic sources such as chloride (**38**), bromide (**39**) or iodide (**40**) resulted in *anti*-halogenated products. Interestingly (Fig. 3), interception with amides such as *N,N*-dimethylacetamide or -formamide resulted in the isolation of *anti*-acyloxy-acylated products (**41** and **42**) whereas—even more intriguingly—the addition of the stable aminoxyl radical TEMPO provided *anti*-OTMP product **43** at high yield. Importantly, when the addition of DMSO was followed by treatment

with triethylamine, the products of a 1,3-ketoacylation reaction were obtained (**44**–**49**). This is a process that, to the best of our knowledge, has no precedent in the literature and converts simple alkenes directly to 1,4-dicarbonyls[37–40].

Aiming to showcase the synthetic prowess of this facile, yet powerful, 1,3-alkene difunctionalization, we explored potential synthetic applications. We selected 4-ipomeanol (**50**), a model pulmonary pretoxin with activity for protein binding (*N*-acetyl cysteine and *N*-acetyl lysine)[41,42], the reported synthesis of which is a multistep endeavour (Fig. 4a; five steps from diethyl furan-3,4-dicarboxylate)[43]. Alkene 1,3-difunctionalization, as presented here, instead enables the one-step synthesis of this compound (as well as the known phenyl-analogue **51** and other derivatives (**52** and **53** (ref. 44)) from inexpensive 1-butene in a straightforward manner.

From a mechanistic point of view (Fig. 4b) we believe that the 1,3-difunctionalizations presented above rely on a rapid isomerization event. This converts, under thermodynamic control, what would be the first intermediate of electrophilic addition[23,24,27], the β-keto cation, into the rearranged, cyclic oxocarbenium ion ***rac*-I**[45–50]—with the formation of ***rac*-I** constituting a locking event to prevent further isomerization and non-selective product formation. This common intermediate is then intercepted either in hydrolytic fashion at the carbonyl (affording the *syn*-configured products) or through invertive displacement at the secondary *sp*³-centre C3 with other nucleophiles, resulting in the formation of the *anti*-configured products described above.

The concept of charge relocation is best illustrated by the mechanistic experiments depicted in Fig. 4c. We established that, regardless of where the carbocation is initially formed (by bromide abstraction), the overwhelming majority of the material is converted into the expected 1,3-difunctionalized target: the carbocation was drawn to the γ position independently of its nascent state.

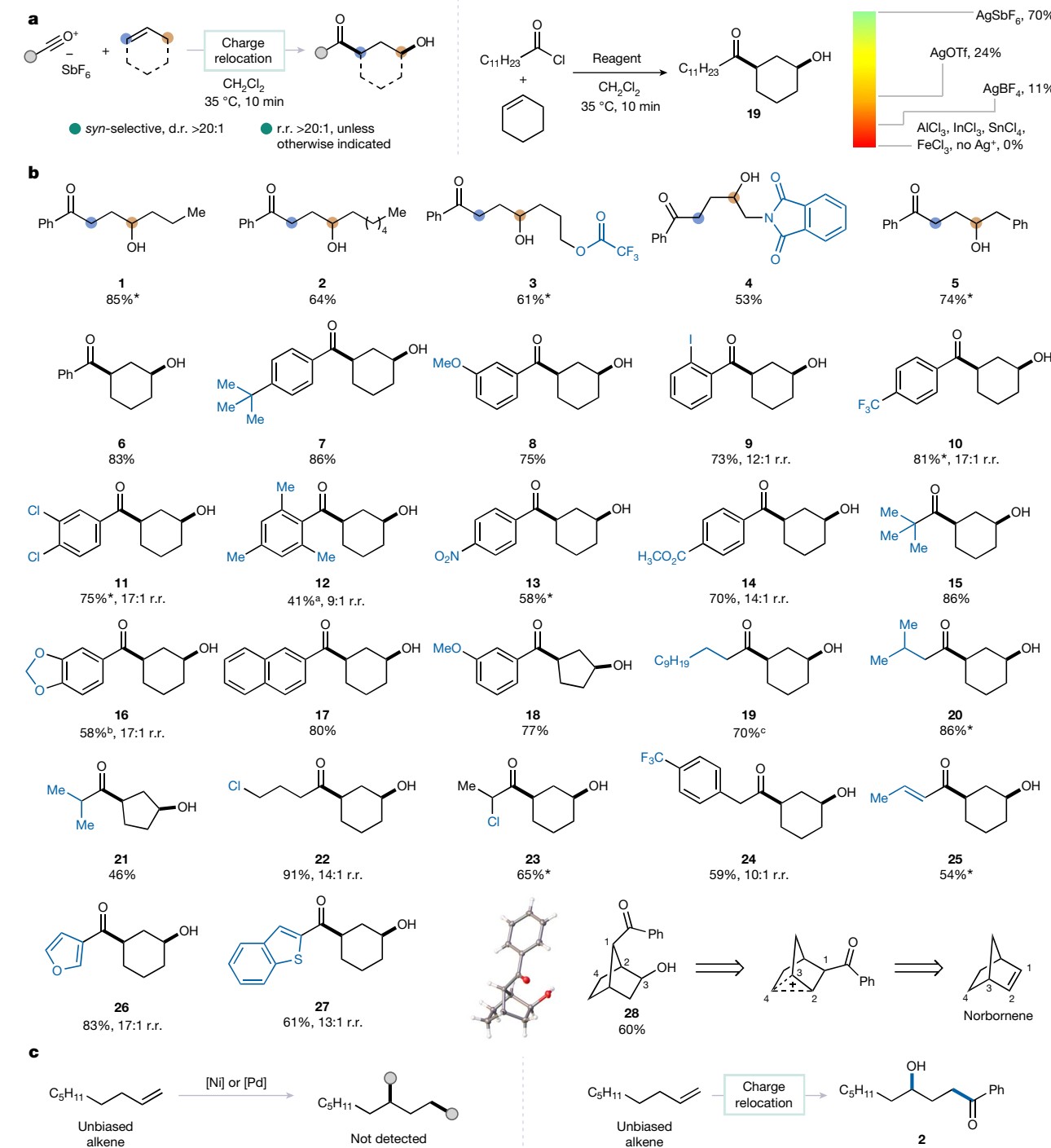

**Fig. 2 | General reaction scheme, optimization and the scope of *syn*-selective 1,3-hydroxyacylation. a**, 1,3-Hydroacylation of cyclohexene. **b**, Scope of *syn*-alcohols. **c**, Comparison with transition-metal catalysis. All yields in the table of optimization correspond to nuclear magnetic resonance yields using 1,3,5-trimethoxybenzene as an internal standard. All products were obtained at greater than 20:1 d.r. and regioisomeric ratio (r.r.), unless otherwise mentioned. All yields correspond to isolated material. See Supplementary Information for further details and additional scope entries. *The reported yields correspond to averages over three runs. In cases in which competing pathways led to the observation of enone byproducts in the crude mixtures, the following percentages were observed: [a]37, [b]24, [c]30. Additional substrate scope is presented in Extended Data Fig. 1 and Supplementary Information.

The reactions of more complex substrates also proved interesting. When 1-methylcyclohexene, a trisubstituted alkene, was used, the all-*syn*-hydroxyketone **54** was obtained as a single diastereomer (see X-ray structure to the right; Fig. 4d). This stereochemical outcome is probably governed by the preferred equatorial orientation of the only substituent not bound within a ring, the methyl group (Fig. 4d).

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

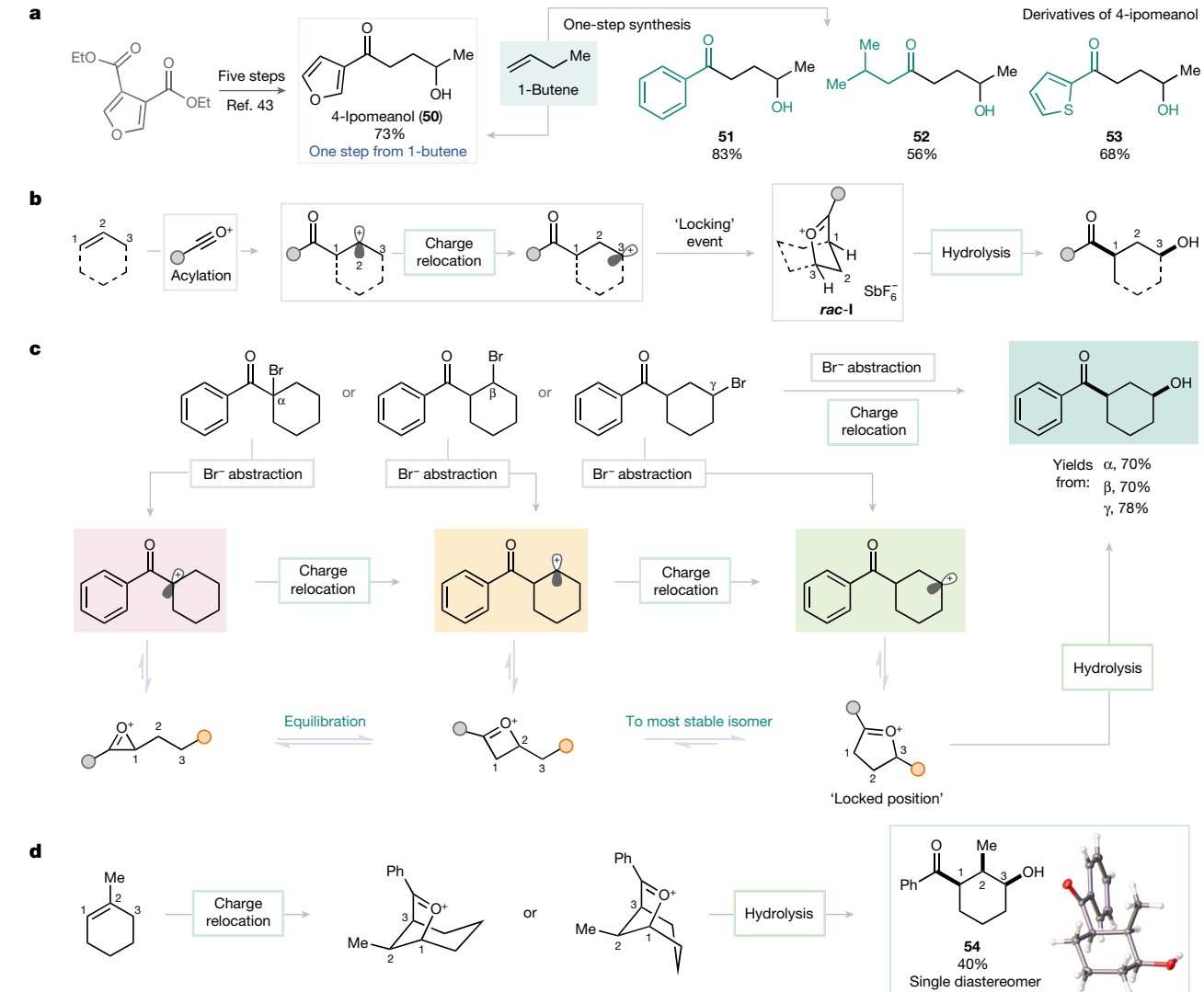

**Fig. 4 | Application and mechanistic investigation of the 1,3-difunctionalization of alkenes. a**, Synthesis of 4-ipomeanol and other biologically active molecules[43]. **b**, Proposed mechanism of 1,3-alkene difunctionalization, involving charge relocation. **c**, Mechanistic investigations

support the hypothesis that nascency of the charge does not affect the constitution of the obtained product (Supplementary Information). **d**, Complete stereocontrol for the formation of a 1,2,3-trisubstituted cyclohexane.

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

# Article

## Data availability

All data are available in the manuscript or Supplementary Information.

**Acknowledgements** We thank A. Roller (University of Vienna) for X-ray crystallographic structure determination. We thank P. S. Grant, C. R. Gonçalves and N. Gillaizeau-Simonian for helpful discussions. This work has been supported by the Austrian Academy of Sciences (DOC Fellowship to B.R.B.) and the European Research Council (CoG VINCAT to N.M.). We thank the University of Vienna for generous support.

**Author contributions** N.M. designed and directed the project. B.R.B., G.I., M.R. and D.K. performed and analysed the experiments. B.R.B., G.I., D.K. and N.M. cowrote the manuscript.

**Competing interests** The authors declare no competing interests.

## Additional information

**Correspondence and requests for materials** should be addressed to Nuno Maulide.

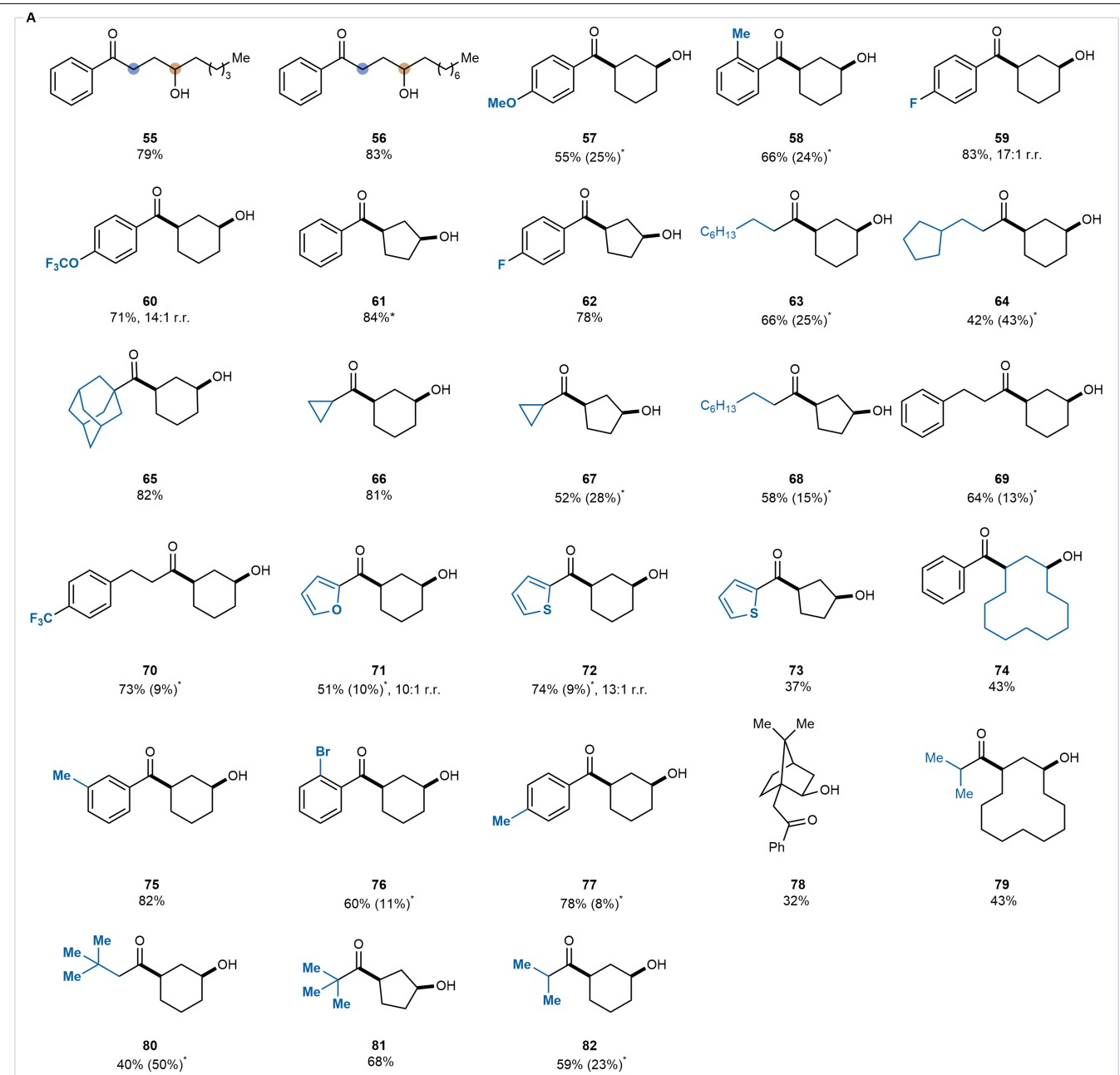

**Extended Data Fig. 1 | Additional products of *syn*-selective 1,3-hydroxyacylation.** *Percentages of observed enone by-products in the crude mixtures (see the Supplementary Information for additional details).