## [Peer Review File · Nature]

Manuscript Title: Stereodivergent 1,3-difunctionalisation of alkenes by charge relocation

Reviewer Comments & Author Rebuttals

Reviewer Reports on the Initial Version:

Referees' comments:

Referee #1 (Remarks to the Author):

In this paper, Maulide and coworkers describe a Friedel–Crafts acylation of cyclic and acyclic unactivated alkenes in which the resulting carbocation intermediate is translocated through intramolecular 1,2-hydride shifts, such that subsequent workup nets overall 1,3-difunctionalization.

Overall, my assessment of this manuscript is mixed. The major negative here is that there is an extensive body of literature on Friedel–Crafts acylation reactions with olefin substrates that is not appropriately contextualized, and well-established reactivity concepts are repackaged and presented as if novel. On the other hand, there are positive aspects to this study. The full synthetic potential of Friedel–Crafts acylation reactions of olefins has not been capitalized on by the synthetic community due to difficulty in controlling the selectivity of the hydride shifts, and the authors have identified conditions that address this problem and shown how it can be applied to the problem of olefin 1,3-difunctionalization, where general solutions are lacking.

The next step should be major revisions and reframing of the introduction such that it rigorously discusses prior art in the field. From here, I am not opposed to reconsideration at Nature, though ultimately, I feel that a more specialized sub-journal, such as Nature Synthesis, will prove to be a better fit.

(1) The authors favor other terminology, but ultimately this reaction is a variant of a classical Friedel–Crafts acylation in which an acylium cation engages with an electron-rich π -system to generate a carbocation intermediate. The Friedel–Crafts acylation of olefins was reviewed as early as 1972 (Chem. Soc. Rev. 1972, 1, 73), and in this review, there is a section devoted to reactions involving subsequent intramolecular hydride transfer. Migration of the carbocation away from the newly installed carbonyl group through hydride shifts due to the destabilizing nature of the proximal electron-withdrawing group has been appreciated since at least the 1960s. It's unclear why the authors choose to coin a new term "charge relocation" to refer to a 1,2-hydride shift in this context. In the introduction, the authors prefer to compare their work to more recent advances in metal-catalyzed migratory alkene functionalization, but this framing completely misses this more relevant body of Friedel–Crafts literature.

(2) While controlling the fate of the carbocation after acylation is generally challenging, there are notable cases where 1,3-difunctionalization has been achieved in high selectivity, as in the case of decalin (which undergoes dehydrogenation and then participates in the same reaction described here). The mechanistic details of this reaction have been extensively studied (J. Am. Chem. 2014,

136, 13745, and references cited therein). Again discussion of this precedent is lacking.

(3) Given that the reactivity itself is not especially new, one of the more interesting aspects of this study is how solvent choice and presumably the non-coordinating counteranion, SbF_6^- , lead to high selectivity for the 1,3-difunctionalized product, as opposed to the α,β -unsaturated olefin (see page S3 of the Supporting Info). This is perhaps the most novel aspect of the study, but it is relegated to the Supporting Info, and no information about other silver salts or other halophilic Lewis acids that were tested is included. Many different Lewis acids have been previously used in olefin Friedel–Crafts acylation so comparing how other classical Lewis acids perform in terms of reactivity and selectivity would be revealing.

(4) Important details of the reaction, including means by which the acylium are generated (acyl chloride + AgSbF_6) and the fact that the method is a two-step telescoped sequence involving a basic workup cannot be found anywhere in the main text. As it stands, the reader must consult the SI to understand the most basic aspects of the chemistry described. Likewise, it is not immediately evident that the anti procedure shown in Figure 3 is in fact a 4-step telescoped sequence.

Referee #2 (Remarks to the Author):

In this manuscript, the authors detail a general and synthetically useful methodology for the 1,3-difunctionalization of unactivated alkenes via a charge relocation pathway. Notably, the difunctionalization can be stereodivergently accomplished with excellent regioselectivity and without the need for directing groups or stabilizing features on the substrate. The authors demonstrate the methodology on a variety of different unactivated alkenes, with alterations of both the electrophiles and nucleophiles used in the transformation, showcasing its ability to access an array of synthetically useful compounds. Moreover, the authors include a brief synthesis of a natural product and its derivatives, as well as mechanistic studies that give evidence to a charge relocation pathway as the operative mode of reactivity.

Due to the simplistic design of the transformation and its ability to provide a general solution to a long-standing challenge in organic synthesis, I believe this study is worthy of publication in *Nature*. Some revisions are suggested, related to prior studies involving the charge relocation pathway for the 1,3-difunctionalization of non-activated alkenes, along with edits to the experimental section.

I recommend further discussion of the following citations in the manuscript:

1. In Ref. 40 of the manuscript, a reaction is reported (termed the “Baddeley Reaction of Decalin”) which is proposed to go through very similar reactive intermediates (acylium cation and unactivated alkene) to those postulated in the current manuscript. The original observation of the reaction was in 1959 by Baddeley and co-workers, see: *Proc. Chem. Soc.* 1959, 395.

2. A report from 1979 demonstrates the reaction of phenylacetyl chloride with cyclohexene in the presence of a Lewis acid, where products are isolated that would arise from a charge relocation pathway see: *J. Org. Chem.* 1979, 44, 1502–1508.

3. A recent OPRD article from 2020 reports the reaction between an acid chloride and cyclopentene, forming a 1,3-difunctionalized side product via a charge relocation pathway. Furthermore, computational analysis was performed to provide for rationale for the observed side product, see: *Org. Process Res. Dev.* 2020, 24, 555–566.

Note: the generality of the transformation developed by Maulide and co-workers is excellent, whereas most of the reports shown above are limited and lower yielding examples.

Specific suggestions are as follows, some of which are very minor:

Manuscript:

- The two categories that the authors define as modes of alkene functionalization (line 24) seem to have overlap. For example, a transformation that results in temporary C-C bond cleavage (defined as category “a”) could also maintain its original C-C sigma-bond connectivity (defined as category “b”). A definition or example of what the authors mean by a temporary C-C bond cleavage could help clarify this for the reader.

- For Ref. 2, if the authors are citing a particular section of the book, it would be best to include the page range. Additionally, an author is missing from the reference, Stuart Warren (it looks like his initials are there, but his surname should be spelled out).

- Does a 1,3-difunctionalization of an alkene, where the alkene functional group is maintained in the product formed from the transformation fall under the categories described in the introduction? If so, studies such as a 1,3-difunctionalization of unactivated alkenes developed by White and co-workers should be cited: *J. Am. Chem. Soc.* 2006, 128, 15076-15077. The citation could be added at line 34. This citation could need clarification in the footnote, as this transformation does not fall under the categories of needing a directing group or “stopper” on the alkene to achieve the desired transformation.

- Whereas trifunctionalizations of alkenes are distinctly different from the reported transformation, they can lead to similar 1,3-difunctionalized products, such as a study from Ge and co-workers where Co-mediated triborylation of unactivated alkenes was achieved and the corresponding 1,1,3-triborylated products could be transformed to the 1,3-diboryl substituted derivative in the presence of base, see: *Angew. Chem., Int. Ed.* 2022, 61, e202116133. This could be important to cite in the manuscript, potentially at line 35 with clarification in the footnote.

- In Figure 1 when showing the Heck-type transformation, it could be useful to have a hypothetical reagent that the grey ball would arise from in the disfavored product. This could be accomplished by having another reagent present over the reaction arrow with the metal-red ball species. This would also be useful for the section below discussing distal functionalization strategies, where the grey ball appears in the products.

- It may be useful to have control experiments present (either in the main text or SI) that provide

evidence that the acylium cation is the reactive electrophile in the transformation. This could be accomplished by performing the reaction in the absence of the silver salt and seeing if product formation occurs. Otherwise, relevant literature should be cited at line 49, that demonstrates that the acylium cation is formed under these reaction conditions (Ag salt or lewis acid and acid chloride)

- Minor typo in Figure 2, in the top right box, "idicated" should be "indicated"

- At line 84, the references after "stereodivergent manner" need to be double checked that they are in the correct location, as it seems that not all of these references belong here. For example, ref. 22 does not seem to comment on stereodivergence.

Experimental (SI)

- For all NMR data reported, please try to remove any solvent or grease and reacquire spectra

- Please specify what DEPT experiment was used for the ^{13}C NMR spectra or give an explanation for the observed phasing. Additionally, whatever phasing or experiments were used should be consistent throughout the SI. For example, the spectrum for compound 1.48 does not have this type of phasing.

- The following ^1H NMR spectra have unlabeled and untabulated peaks that need further explanation: 1.01, 1.02, 1.05, 1.16, 1.19, 1.23, 1.27, 1.28, 1.34, 1.38, 1.41, 1.46, 1.50, 2.01, 2.02, 2.10, 3.02, 4.02. Otherwise, cleaner spectra need to be provided.

- The following ^{13}C NMR spectra have unlabeled and untabulated peaks that need further explanation: 1.02, 1.04, 1.05, 1.07, 1.10, 1.15, 1.26-1.28, 1.29 (extraneous peaks are picked but not in the tabulated data), 1.30, 1.34, 1.35, 1.36 (extraneous peaks are picked but not in the tabulated data), 1.37, 1.41, 2.01, 2.02, 2.04, 2.06, 4.01, 4.06. If these extraneous peaks are a minor isomer, it would be helpful to specify on each spectrum this is the case (despite the disclaimer at the beginning of the section, it can be hard to tell). Otherwise, cleaner spectra need to be provided.

- The following ^{13}C NMR spectra have unlabeled and untabulated peak at 33 ppm, with peaks sometimes present at 67 ppm and 22 ppm as well: 1.11-1.14, 1.16, 1.18, 1.19, 1.21, 1.22, 1.32, 1.38-1.40, 1.42-1.45, 1.49-1.51, 1.54, 1.56, 2.05, 2.07, 2.09, 4.02. Otherwise, cleaner spectra need to be provided.

- It is important to provide specific 2D NMR correlations and spectra or relevant literature that demonstrates how relative stereochemistry was determined for the diastereomers.

- For compound S4, it looks like some of the carbon peaks in the ^{13}C NMR spectrum have been peak picked twice, as they have either two identical or slightly different values. Additionally the values of the picked peaks in the spectrum are off from the tabulated data by ~ 0.15 ppm.

- For general procedures 2, 3, 4, and 5, are these reactions performed under inert atmosphere? Please adapt the procedures to specify.

- For general procedures, 2, 3, 4, and 5, please include the volumes used for the workup of the reaction (i.e. the amount of DCM used for each extraction and the total number of extractions)
- At line 104, please include how the reaction is able to stir in the procedure (i.e. oven-dried vial charged with a magnetic stir bar). This should be adapted to the other general procedures as well.
- At line 109, was the reaction cooled to room temperature before adding the solution of sodium bicarbonate? This should be adapted to the other general procedures as well.
- How many times were the isolation experiments replicated for each substrate? Would be good to see that the transformation is consistently reproducible.
- For all characterized compounds, please include R_f values, as well as melting points (when the product is a solid).
- For compound 1.04, the ¹³C NMR spectrum has an additional proton integrated, most likely in the 1.14–1.39 ppm region
- For compound 1.25, there are only 10 carbons accounted for in the peaks tabulated for the ¹³C NMR spectrum, and there should be 12. Additionally, compound 1.39 only has 15 carbons accounted for in the peaks tabulated, and there should be 16.
- At line 644, instead of “15”, it should be “15H”
- Protons are missing from the tabulated ¹H NMR data and spectra of 1.21 and 1.32.
- For compound 1.33, if the carbonyl peak in the tabulation for the ¹³C NMR is “seen only in HMBC” can you please provide the relevant HMBC correlations and spectrum that provides evidence for this? This also applies to compound 3.06, as many of the carbon peaks tabulated are “seen only by HMBC or HSQC”.
- There are two extra protons tabulated for the ¹H NMR spectrum of compound 1.35.
- Please provide relevant NMR spectra for compounds S5, S6, S7, S9, S10, S11, and S12

Author Rebuttals to Initial Comments:

Referees' comments:

Referee #1 (Remarks to the Author):

In this paper, Maulide and coworkers describe a Friedel–Crafts acylation of cyclic and acyclic unactivated alkenes in which the resulting carbocation intermediate is translocated through intramolecular 1,2-hydride shifts, such that subsequent workup nets overall 1,3-difunctionalization.

Overall, my assessment of this manuscript is mixed. The major negative here is that there is an extensive body of literature on Friedel–Crafts acylation reactions with olefin substrates that is not appropriately contextualized, and well-established reactivity concepts are repackaged and presented as if novel. On the other hand, there are positive aspects to this study. The full synthetic potential of Friedel–Crafts acylation reactions of olefins has not been capitalized on by the synthetic community due to difficulty in controlling the selectivity of the hydride shifts, and the authors have identified conditions that address this problem and shown how it can be applied to the problem of olefin 1,3-difunctionalization, where general solutions are lacking.

The next step should be major revisions and reframing of the introduction such that it rigorously discusses prior art in the field. From here, I am not opposed to reconsideration at Nature, though ultimately, I feel that a more specialized sub-journal, such as Nature Synthesis, will prove to be a better fit.

(1) The authors favor other terminology, but ultimately this reaction is a variant of a classical Friedel–Crafts acylation in which an acylium cation engages with an electron-rich π -system to generate a carbocation intermediate. The Friedel–Crafts acylation of olefins was reviewed as early as 1972 (Chem. Soc. Rev. 1972, 1, 73), and in this review, there is a section devoted to reactions involving subsequent intramolecular hydride transfer. Migration of the carbocation away from the newly installed carbonyl group through hydride shifts due to the destabilizing nature of the proximal electron-withdrawing group has been appreciated since at least the 1960s. It's unclear why the authors choose to coin a new term “charge relocation” to refer to a 1,2-hydride shift in this context. In the introduction, the authors prefer to compare their work to more recent advances in metal-catalyzed migratory alkene functionalization, but this framing completely misses this more relevant body of Friedel–Crafts literature.

We thank the reviewer for their valuable input and suggestion. Indeed, we entirely agree that this important body of work should not and cannot be omitted from the introduction and is worthy of discussion, in order to place our work into the appropriate context. We have therefore added a paragraph discussing the general reactivity of the Friedel-Crafts reaction of alkenes and we have added Figure 1B, highlighting important precedent of similar transformations and providing context for the significance of our work.

We hope that the newly provided context also helps clarify our choice of terminology. “Charge relocation” conceptualises a type of reactivity that may have been previously observed, but which was never as (a) general, (b) predictable and (c) selective as we report in this manuscript.

(2) While controlling the fate of the carbocation after acylation is generally challenging, there are notable cases where 1,3-difunctionalization has been achieved in high selectivity, as in the case of decalin (which undergoes dehydrogenation and then participates in the same reaction described here). The mechanistic details of this reaction have been extensively studied (J. Am. Chem. 2014, 136, 13745, and references cited therein). Again discussion of this precedent is lacking.

The Baddeley reaction indeed deserves more credit, as one of the earliest—and in fact one of the only—examples of regio- and stereoselective 1,3-difunctionalisation reactions of alkenes using acylium ions. We have therefore included it in both the introduction and in Figure 1B.

(3) Given that the reactivity itself is not especially new, one of the more interesting aspects of this study is how solvent choice and presumably the non-coordinating counteranion, SbF₆⁻, lead to high selectivity for the 1,3-difunctionalized product, as opposed to the alpha,beta-unsaturated olefin (see page S3 of the Supporting Info). This is perhaps the most novel aspect of the study, but it is relegated to the Supporting Info, and no information about other silver salts or other halophilic Lewis acids that were tested is included. Many different Lewis acids have been previously used in olefin Friedel–Crafts acylation so comparing how other classical Lewis acids perform in terms of reactivity and selectivity would be revealing.

We thank the reviewer for pointing out the benefits of highlighting these findings in the main text. Additional information regarding optimisation/the results obtained under different conditions (including other silver salts, halophilic Lewis acids and reaction in the absence of silver) has been added to both the results and discussion section and to Figure 2. Notably, limitations with respect to the figure size and word count mandate the inclusion of additional results (e.g. reaction outcome using different solvents) only in the Supporting Information.

(4) Important details of the reaction, including means by which the acylium are generated (acyl chloride + AgSbF₆) and the fact that the method is a two-step telescoped sequence involving a basic workup cannot be found anywhere in the main text. As it stands, the reader must consult the SI to understand the most basic aspects of the chemistry described. Likewise, it is not immediately evident that the anti procedure shown in Figure 3 is in fact a 4-step telescoped sequence.

We agree with the reviewer's assessment regarding the details of the synthetic transformation. In order to provide the reader with more immediate information, we have highlighted the means of acylium generation both in the main text and in a new panel within Figure 2 (showing that the main method consists of a one-step transformation with a work-up), which shows some results of the optimisation (the entirety of which can be found in the SI). We have also now indicated in both the main text and in Figure 4B and C that hydrolytic work-up is required before obtention of the product. Additionally, in the caption to Figure 3 we now detail more clearly the operations involved towards the respective products.

Referee #2 (Remarks to the Author):

In this manuscript, the authors detail a general and synthetically useful methodology for the 1,3-difunctionalization of unactivated alkenes via a charge relocation pathway. Notably, the difunctionalization can be stereodivergently accomplished with excellent regioselectivity and without the need for directing groups or stabilizing features on the substrate. The authors demonstrate the methodology on a variety of different unactivated alkenes, with alterations of both the electrophiles and nucleophiles used in the transformation, showcasing its ability to access an array of synthetically useful compounds. Moreover, the authors include a brief synthesis of a natural product and its derivatives, as well as mechanistic studies that give evidence to a charge relocation pathway as the operative mode of reactivity.

Due to the simplistic design of the transformation and its ability to provide a general solution to a long-standing challenge in organic synthesis, I believe this study is worthy of publication in Nature. Some revisions are suggested, related to prior studies involving the charge relocation pathway for the 1,3-difunctionalization of non-activated alkenes, along with edits to the experimental section.

I recommend further discussion of the following citations in the manuscript:

1. In Ref. 40 of the manuscript, a reaction is reported (termed the “Baddeley Reaction of Decalin”) which is proposed to go through very similar reactive intermediates (acylium cation and unactivated alkene) to those postulated in the current manuscript. The original observation of the reaction was in 1959 by Baddeley and co-workers, see: Proc. Chem. Soc. 1959, 395.

We thank the reviewer for pointing out the omission of the original work. The reference has been included and we have added the Baddeley to the introduction and Figure 1B.

2. A report from 1979 demonstrates the reaction of phenylacetyl chloride with cyclohexene in the presence of a Lewis acid, where products are isolated that would arise from a charge relocation pathway see: J. Org. Chem. 1979, 44, 1502–1508.

We thank the reviewer for pointing out this highly pertinent reference. We have included it both in the introduction and in Figure 1B, where we highlight the two products resulting from a relocation of the positive charge.

3. A recent OPRD article from 2020 reports the reaction between an acid chloride and cyclopentene, forming a 1,3-difunctionalized side product via a charge relocation pathway. Furthermore, computational analysis was performed to provide for rationale for the observed side product, see: Org. Process Res. Dev. 2020, 24, 555–566.

We thank the reviewer for alerting us to this message. In fact, on the day that we received the reviewers' reports, the same research group published an additional study, in which a 1,3-hydroxyacylation product was identified as a minor component of the reported reaction. We have therefore added both the OPRD2020, as well as the follow up study (and others), to provide proper context.

Note: the generality of the transformation developed by Maulide and co-workers is excellent, whereas most of the reports shown above are limited and lower yielding examples.

Specific suggestions are as follows, some of which are very minor:

Manuscript:

- The two categories that the authors define as modes of alkene functionalization (line 24) seem to have overlap. For example, a transformation that results in temporary C-C bond cleavage (defined as category “a”) could also maintain its original C-C sigma-bond connectivity (defined as category “b”). A definition or example of what the authors mean by a temporary C-C bond cleavage could help clarify this for the reader.

We appreciate the reviewer’s comment and food for thought. In order to circumvent ambiguity or overlap in the definitions, we have added examples for the temporary (metathesis) or permanent (ozonolysis) cleavage and have specified this cleavage as pertaining to the C=C double bond, rather than simply writing C–C bond.

- For Ref. 2, if the authors are citing a particular section of the book, it would be best to include the page range. Additionally, an author is missing from the reference, Stuart Warren (it looks like his initials are there, but his surname should be spelled out).

We thank the reviewer for this valuable comment. We have corrected the reference.

- Does a 1,3-difunctionalization of an alkene, where the alkene functional group is maintained in the product formed from the transformation fall under the categories described in the introduction? If so, studies such as a 1,3-difunctionalization of unactivated alkenes developed by White and co-workers should be cited: J. Am. Chem. Soc. 2006, 128, 15076-15077. The citation could be added at line 34. This citation could need clarification in the footnote, as this transformation does not fall under the categories of needing a directing group or “stopper” on the alkene to achieve the desired transformation.

We thank the reviewer for pointing out this report. We have added the reference following the sentence “while the nascent field of remote functionalisation has emerged as a way of redirecting reactivity towards a distal position, away from the initial reactive site”. Here, we believe no further clarifying statement (which we could not have added as a footnote due to the Nature formatting guidelines) is necessary.

- Whereas trifunctionalizations of alkenes are distinctly different from the reported transformation, they can lead to similar 1,3-difunctionalized products, such as a study from Ge and co-workers where Co-mediated triborylation of unactivated alkenes was achieved and the corresponding 1,1,3-triborylated products could be transformed to the 1,3-diboryl substituted derivative in the presence of base, see: Angew. Chem., Int. Ed. 2022, 61, e202116133. This could be important to cite in the manuscript, potentially at line 35 with clarification in the footnote.

Similarly, we have added this transformation to the main text and thank the reviewer for pointing it out.

- In Figure 1 when showing the Heck-type transformation, it could be useful to have a hypothetical reagent that the grey ball would arise from in the disfavored product. This could be accomplished by having another reagent present over the reaction arrow with the metal-red ball species. This would also be useful for the section below discussing distal functionalization strategies, where the grey ball appears in the products.

We fully agree with the reviewer that additional information would serve the reader well in helping to understand the type of reactivity that we describe in Figure 1A. We have therefore added this information.

- It may be useful to have control experiments present (either in the main text or SI) that provide evidence that the acylium cation is the reactive electrophile in the transformation. This could be accomplished by performing the reaction in the absence of the silver salt and seeing if product formation occurs. Otherwise, relevant literature should be cited at line 49, that demonstrates that the acylium cation is formed under these reaction conditions (Ag salt or Lewis acid and acid chloride)

We agree with the reviewer. Their comment, in combination with a point brought up by reviewer 1, has led us to include an overview of different silver salts, a range of halophilic Lewis acids and a reaction in the absence of either in Figure 2A.

- Minor typo in Figure 2, in the top right box, “idicated” should be “indicated”

We thank the reviewer for their attention to detail. We have corrected this mistake.

- At line 84, the references after “stereodivergent manner” need to be double checked that they are in the correct location, as it seems that not all of these references belong here. For example, ref. 22 does not seem to comment on stereodivergence.

We thank the reviewer for their attention to detail. We have addressed this issue and all references have been placed in the appropriate position within the manuscript.

Experimental (SI)

- For all NMR data reported, please try to remove any solvent or grease and reacquire spectra.

We have done this to the best of our ability.

- Please specify what DEPT experiment was used for the ¹³C NMR spectra or give an explanation for the observed phasing. Additionally, whatever phasing or experiments were used should be consistent throughout the SI. For example, the spectrum for compound 1.48 does not have this type of phasing.

The ¹³C NMR spectra have been unified. Only in instances where particular signals were only (or better) visible in APT mode, have these spectra been added additionally to the DEPT spectra.

- The following ¹H NMR spectra have unlabeled and untabulated peaks that need further explanation: 1.01, 1.02, 1.05, 1.16, 1.19, 1.23, 1.27, 1.28, 1.34, 1.38, 1.41, 1.46, 1.50, 2.01, 2.02, 2.10, 3.02, 4.02. Otherwise, cleaner spectra need to be provided.

We thank the reviewer for pointing this out. In many cases, the additional peaks correspond to the minor regioisomer – in those cases, it has been indicated as such. In the other cases, the compounds have been repurified, yields recalculated and new spectra have been added.

- The following ¹³C NMR spectra have unlabeled and untabulated peaks that need further explanation: 1.02, 1.04, 1.05, 1.07, 1.10, 1.15, 1.26-1.28, 1.29 (extraneous peaks are picked but not in the tabulated data), 1.30, 1.34, 1.35, 1.36 (extraneous peaks are picked but not in the tabulated data), 1.37, 1.41, 2.01, 2.02, 2.04, 2.06, 4.01, 4.06. If these extraneous peaks are a minor isomer, it would be helpful to specify on each spectrum this is the case (despite the disclaimer at the beginning of the section, it can be hard to tell). Otherwise, cleaner spectra need to be provided.

We thank the reviewer for pointing this out. In some cases, the additional peaks correspond to the minor regioisomer – in those cases, it has been indicated as such. In the other cases, the compounds have been repurified, yields recalculated and new spectra have been added.

- The following ¹³C NMR spectra have unlabeled and untabulated peak at 33 ppm, with peaks sometimes present at 67 ppm and 22 ppm as well: 1.11-1.14, 1.16, 1.18, 1.19, 1.21, 1.22, 1.32, 1.38-1.40, 1.42-1.45, 1.49-1.51, 1.54, 1.56, 2.05, 2.07, 2.09, 4.02 . Otherwise, cleaner spectra need to be provided.

We thank the reviewer for pointing this out. In many cases, the additional peaks correspond to the minor regioisomer – in those cases, it has been indicated as such. In the other cases, the compounds have been repurified, yields recalculated and new spectra have been added.

- It is important to provide specific 2D NMR correlations and spectra or relevant literature that demonstrates how relative stereochemistry was determined for the diastereomers.

We have added the NOESY spectrum of a representative structure. Together with the obtained X-ray structures, this provides additional support for the reported relative stereochemistry.

- For compound S4, it looks like some of the carbon peaks in the ¹³C NMR spectrum have been peak picked twice, as they have either two identical or slightly different values. Additionally the values of the picked peaks in the spectrum are off from the tabulated data by ~0.15 ppm.

We thank the reviewer for pointing this out. This has been fixed.

- For general procedures 2, 3, 4, and 5, are these reactions performed under inert atmosphere? Please adapt the procedures to specify.

We have added this information to the general procedures.

- For general procedures, 2, 3, 4, and 5, please include the volumes used for the workup of the reaction (i.e. the amount of DCM used for each extraction and the total number of extractions)

We have added this information to the general procedures.

- At line 104, please include how the reaction is able to stir in the procedure (i.e. oven-dried vial charged with a magnetic stir bar). This should be adapted to the other general procedures as well.

We have added this information to the general procedures.

- At line 109, was the reaction cooled to room temperature before adding the solution of sodium bicarbonate? This should be adapted to the other general procedures as well.

The solution of sodium bicarbonate was added immediately after removal of the vial from the oil bath. We have added this information to the general procedures.

- How many times were the isolation experiments replicated for each substrate? Would be good to see that the transformation is consistently reproducible.

We have added information to the main text regarding the reproducibility of selected examples. While not all examples were performed multiple times, several products were obtained in very consistent yields on several occasions; where this is the case, this has been indicated.

- For all characterized compounds, please include R_f values, as well as melting points (when the product is a solid).

We have added this information to the general procedures.

- For compound 1.04, the ¹³C NMR spectrum has an additional proton integrated, most likely in the 1.14–1.39 ppm region.

We thank the reviewer for their attention to detail. This has been fixed.

- For compound 1.25, there are only 10 carbons accounted for in the peaks tabulated for the ¹³C NMR spectrum, and there should be 12. Additionally, compound 1.39 only has 15 carbons accounted for in the peaks tabulated, and there should be 16.

We thank the reviewer for their attention to detail. This has been fixed.

- At line 644, instead of “15”, it should be “15H)”

We thank the reviewer for their attention to detail. This has been fixed.

- Protons are missing from the tabulated ¹H NMR data and spectra of 1.21 and 1.32.

We thank the reviewer for their attention to detail. This has been fixed.

- For compound 1.33, if the carbonyl peak in the tabulation for the ¹³C NMR is “seen only in HMBC” can you please provide the relevant HMBC correlations and spectrum that provides evidence for this? This also applies to compound 3.06, as many of the carbon peaks tabulated are “seen only by HMBC or HSQC”.

We have been able to obtain a new ^{13}C NMR spectrum for compound 1.33, in which the corresponding carbonyl peak can be seen. For compound 3.06, we have added the corresponding HMBC and HSQC correlations in the NMR section.

- There are two extra protons tabulated for the ^1H NMR spectrum of compound 1.35.

We thank the reviewer for their attention to detail. This has been fixed.

- Please provide relevant NMR spectra for compounds S5, S6, S7, S9, S10, S11, and S12

We have added the corresponding NMR spectra. Please note that for some of these compounds, in full accordance with the cited literature, we were not able to obtain entirely pure spectra. However, the spectroscopic data obtained unambiguously confirm the nature of the products.

Reviewer Reports on the First Revision:

Referees' comments:

Referee #1 (Remarks to the Author):

Overall the revised version of the manuscript is much improved in terms of scholarship. In particular, the authors have added relevant references and addressed several issues in the supporting information. The authors added one paragraph and accompanying figures to discuss prior work on Friedel–Crafts acylation of olefins in response to referee feedback.

Because my concerns vis-à-vis limited novelty have not been entirely allayed, I would still favor publication in a more specialized journal. Issues to address are listed below, beyond which I do not have a strong objection to publication at this stage.

(1) A lingering concern is that it's not entirely clear to what extent the authors have actually solved the selectively control problem in cation migration. The selectivity for 1,3-product to enone with the standard substrate, cyclohexene, is only 2.3:1, though this data is relegated to the Supporting Information (p. S2). It's unclear how high or low these ratios are with other substrates since this point is not commented on.

Please note that in the originally submitted version of the SI, this key experiment in the optimization table was reported to not yield any detectable byproduct (70% product yield, enone byproduct not detected). While I appreciate the authors updating the data to reflect current knowledge most accurately, the updated table nevertheless gives rise to the concerns about selectivity.

(2) Generally speaking, I am not a fan of the term "charge relocation", which I view as effectively a rebranding of sequential hydride shifts. I will defer to the editor on this matter, but in any case, if the language is kept, I recommend that term be defined upon first use in the main text (at the end of page 2, paragraph 2).

(3) In Fig 1, panel A: for the depiction of nickel-catalyzed 1,3-difunctionalization, I would recommend that it be specified that the Heck byproduct is a minor product in the systems referred to here, either by using the words "major/minor" or fading out the Heck byproduct.

(4) Fig 1, panel B, top: I think it would be clearer how this product distribution arises if the authors were to draw the initially formed carbocation intermediate.

(5) In Fig 1, panel B, bottom: the authors show Δ 9,10-octalin and report the result as "yield not reported". This strikes me as a disingenuous description since the yield of the exact experiment is reported in one of the references provided during the last round of peer review (J. Am. Chem. Soc. 2014, 136, 13747): 32% of the non-hydrolyzed product.

(6) In the revised version of Figure 2, it would be instructive to show the yield of desired product and major byproduct (enone) (and product ratio) to depict how selectivity changes across different

counteranions (data available on page S2 in the Supporting Information). Note that the selectivity of desired to undesired under optimal conditions is only 2.3:1

(7) In the mechanistic discussion, I recommend that relevant papers from the Friedel–Crafts literature (now cited and discussed in the introduction) be cited here to clarify that the general mechanistic features of this hydride shift / trapping process have been previously appreciated and elucidated.

(8) Some minor typographical issues are listed below:

- Page 1, paragraph 1: “no less than” -> “no fewer than”
- Page 2, paragraph 1: insert comma before “and the direct”
- Page 3, figure 1 and caption “Friedel–Crafts” (en-dash)
- Figures 2 and 3: there may have been an issue with the PDF conversion, because the text boxes do not appear properly aligned throughout these figures. For example the solvent and temperature are not aligned in Figure 2, panel A

Referee #2 (Remarks to the Author):

I have reviewed all of the reviewer comments and think the authors have done an excellent job with this manuscript. I think publication in Nature is well warranted and I expect this manuscript will be well received by the community.

The manuscript does not need further revision, but I would suggest minor revisions to the experimental document:

Please double-check the tabulation for compound 1.32's NMR data. I think there is an extra proton.

Some of the spectra are a bit messy, as the authors mention. I will leave it to the journal as to whether or not to require spectra of purified materials, as this may be important to the journal.

Author Rebuttals to First Revision:

Referees' comments:

Referee #1 (Remarks to the Author):

Overall the revised version of the manuscript is much improved in terms of scholarship. In particular, the authors have added relevant references and addressed several issues in the supporting information. The authors added one paragraph and accompanying figures to discuss prior work on Friedel–Crafts acylation of olefins in response to referee feedback.

We thank the reviewer for acknowledging our efforts to improve the manuscript, and we remain grateful for the constructive input provided.

Because my concerns vis-à-vis limited novelty have not been entirely allayed, I would still favor publication in a more specialized journal. Issues to address are listed below, beyond which I do not have a strong objection to publication at this stage.

(1) A lingering concern is that it's not entirely clear to what extent the authors have actually solved the selectively control problem in cation migration. The selectivity for 1,3-product to enone with the standard substrate, cyclohexene, is only 2.3:1, though this data is relegated to the Supporting Information (p. S2). It's unclear how high or low these ratios are with other substrates since this point is not commented on.

We thank the reviewer for this comment and wish to present a clarification. Indeed, we understand that the substrate shown for optimisation could evoke the feeling that the method presented in this manuscript lacks general selectivity for the 1,3-difunctionalisation.

Initially, it is worthy of note that the vast majority of reactions did not show any enone formation and among those that did, the standard reactant combination used for optimisation (lauroyl chloride + cyclohexene) provided one of the lowest ratios of product/alkene formation (only six other reactions gave lower selectivity). Moreover, we also want to highlight that in the reactant combination chosen for optimisation, it was the acyl chloride, not the alkene (no general trend or propensity to give elimination products was observed for any of the variety of alkenes tested), which provoked by-product formation. We would like to point out that we deliberately chose a more challenging combination of reagents for the optimisation, so as to provide the best achievable and most general reaction conditions. This is important, as it shows that the selectivity is not at all alkene (i.e. substrate) dependent, but the selectivity for product distribution can be influenced by choice of the electrophilic reagent.

This can be reinforced by analysing product distribution for all reactions shown:

Following analysis of the crude NMR spectra of all 83 examples, we found the following:

63 examples: no enone observed (100% selectivity)

2 examples: 90-99% selectivity for the desired product

6 examples: 80-89% selectivity for the desired product

6 examples: 70-79% selectivity for the desired product

6 examples: <70% selectivity for the desired product

Mean selectivity: 94% (corresponds to an average ratio of 16:1)

We have provided an empirical rationale for enone formation in the SI:

When enone formation is observed, it is almost exclusively for products derived from acyl chlorides that are either comparatively electron rich or aliphatic. Thus, it could be surmised that the basicity of the carbonyl (being greater for electron-rich carbonyls) leads to increased levels of deprotonation. However, examples such as 1.29 and 1.33 do not entirely agree with the observed trend and therefore we refrain from making a conclusive statement in the manuscript itself.

So as to not overload Figures 2 and 3, as well as the corresponding captions, we chose to report selectivity ranges (as shown above) in the captions. We furthermore wish to highlight that, in all cases, any enone byproducts were easily separable from the charge relocation products.

In order to provide a full picture, we have, however, included annotated versions of Figures 2 and 3 in the supporting information (section 2.1).

In summary:

- i) the reactant combination chosen for optimisation is not representative of a general trend; it was the acyl chloride, not the alkene, which provoked by-product formation;
- ii) we deliberately chose a more challenging combination of reagents for the optimisation, so as to provide the best achievable and most general reaction conditions; and
- iii) iii) the data we have now provided clearly show that the vast majority of examples sees no enone formation at all, and a mean selectivity of 94% across the board is observed.

Please note that in the originally submitted version of the SI, this key experiment in the optimization table was reported to not yield any detectable byproduct (70% product yield, enone byproduct not detected). While I appreciate the authors updating the data to reflect current knowledge most accurately, the updated table nevertheless gives rise to the concerns about selectivity.

We understand that the new information provided in the previously submitted version of the revisions caused concern—but we had deliberately chosen one of the most difficult examples. However, we have clearly shown that this result is to be considered more an “outlier” than a representative example and that selectivity for the desired product is high (this is further illustrated by the median value for product/enone selectivity is 100%).

(2) Generally speaking, I am not a fan of the term “charge relocation”, which I view as effectively a rebranding of sequential hydride shifts. I will defer to the editor on this matter, but in any case, if the language is kept, I recommend that term be defined upon first use in the main text (at the end of page 2, paragraph 2).

We appreciate the reviewer's comment and have defined the term upon first use in the main text. "Charge relocation" is a process by which incipient or localised charge at a given atom is driven, through a device introduced by synthesis, to relocate with high preference and selectivity to a defined position. "Charge relocation" furthermore denotes processes where this can be achieved across a range of substrate/reactant combinations.

(3) In Fig 1, panel A: for the depiction of nickel-catalyzed 1,3-difunctionalization, I would recommend that it be specified that the Heck byproduct is a minor product in the systems referred to here, either by using the words "major/minor" or fading out the Heck byproduct.

We thank the reviewer for pointing this out and have added "major/minor" to the products. We have also added additional information to the text describing this panel, providing an explanation for the formation of the formation of Heck-type by-products.

(4) Fig 1, panel B, top: I think it would be clearer how this product distribution arises if the authors were to draw the initially formed carbocation intermediate.

We have now included this intermediate with a carbocation in beta-position with respect to the newly installed acyl group.

(5) In Fig 1, panel B, bottom: the authors show Δ 9,10-octalín and report the result as "yield not reported". This strikes me as a disingenuous description since the yield of the exact experiment is reported in one of the references provided during the last round of peer review (J. Am. Chem. Soc. 2014, 136, 13747): 32% of the non-hydrolyzed product.

While there is indeed a 32% yield of the non-hydrolyzed product reported, we respectfully disagree that this is to be equated with the yield for the hydrolyzed keto-alcohol. Indeed, the literature (Synlett 2011, 15, 2211 – see screenshots below, from Schemes 2 and 3) suggests that hydrolysis to the corresponding keto-alcohol is not high yielding (51% yield reported; in this report, this equates to an overall yield of 13%). In contrast, none of the prior reports of this reaction specify any direct yields for these types of product.

While the 1959 report does not show a yield, we appreciate the reviewer pointing out the yields reported in later work. We have thus added a yield of 13% (and included the Synlett 2011 reference) to the Figure.

Scheme 2

(6) In the revised version of Figure 2, it would be instructive to show the yield of desired product and major byproduct (enone) (and product ratio) to depict how selectivity changes across different

counteranions (data available on page S2 in the Supporting Information). Note that the selectivity of desired to undesired under optimal conditions is only 2.3:1

As above, we fully agree with the reviewer's assessment that inclusion of information regarding product distribution is warranted. In order to avoid overburdening the already information-heavy Figure 2, we have—as stated above—chosen to include information pertaining to enone formation in the caption. We have done this both for the substrate scope and the survey of different counteranions.

(7) In the mechanistic discussion, I recommend that relevant papers from the Friedel–Crafts literature (now cited and discussed in the introduction) be cited here to clarify that the general mechanistic features of this hydride shift / trapping process have been previously appreciated and elucidated.

We have repeated the references here, in line with the reviewer's recommendation.

(8) Some minor typographical issues are listed below:

- Page 1, paragraph 1: “no less than” -> “no fewer than”
- Page 2, paragraph 1: insert comma before “and the direct”
- Page 3, figure 1 and caption “Friedel–Crafts” (en-dash)
- Figures 2 and 3: there may have been an issue with the PDF conversion, because the text boxes do not appear properly aligned throughout these figures. For example the solvent and temperature are not aligned in Figure 2, panel A

We thank the reviewer for their attention to detail. All changes have been made. We believe the problem with PDF conversion to have arisen from variable/dissimilar line spacing within the chemdraw files – this has now been fixed.

Referee #2 (Remarks to the Author):

I have reviewed all of the reviewer comments and think the authors have done an excellent job with this manuscript. I think publication in Nature is well warranted and I expect this manuscript will be well received by the community.

The manuscript does not need further revision, but I would suggest minor revisions to the experimental document:

Please double-check the tabulation for compound 1.32's NMR data. I think there is an extra proton.

We thank the reviewer for their attention to detail and apologise for the oversight. This has been fixed.

Some of the spectra are a bit messy, as the authors mention. I will leave it to the journal as to whether or not to require spectra of purified materials, as this may be important to the journal.

Reviewer Reports on the Second Revision:

Referees' comments:

Referee #1 (Remarks to the Author):

The authors have made several changes to the manuscript to reflect concerns raised during the previous round of peer review. Most significantly, they now address the issue of enone byproduct formation for every example reported in the paper. To summarize their findings, it is not a general phenomenon and seems to become more pronounced with certain acylium cations (in a manner that is not straightforward to rationalize). The percent enone formation is now reported in the footnotes and in fully annotated versions of the scope table that are included in the SI. The presence of these byproducts is now also discussed in the main text. From my perspective, this issue has been thoroughly addressed, and my concerns are allayed.

Other concerns (more minor in nature) still persist and are listed below:

(1) Regarding the term "charge location", again I will defer to the editor on this matter, but I'm not convinced there is a need for inventing this term to describe a phenomenon that has already been described many times using more conventional and established terminology, hydride shift(s). Do the authors view charge relocation as being a more general phenomenon beyond hydride shifts (and possibly alkyl shifts)?

I don't necessarily find issue with creating a term that the authors think will be more appealing to a broad audience, but then when defined upon first use, it should be clear that in this context, "charge relocation" is taking place via successive hydride shifts. (Note that "hydride shift" is never used in the manuscript, despite being the clearest terminology and the terminology established previously for this exact process.)

(2) The claim that the decalin-derived vinyl ether had never been hydrolyzed with a reported yield until 2011 does not appear to be true.

The same authors as the 1959 paper in question reported follow up studies where yields for this step are also included (see, Baddeley et al., J. Chem. Soc. 1960, 4713). Excerpt reproduced below for convenience.

10p-Acetyl-trans-decal-1(3-ol (VI).—A mixture of the vinyl ether (4.5 g.) in ether (30 ml.) and dilute sulphuric acid (60 ml.; x) was stirred and gently refluxed for 2.5 hr. The ether layer was separated and, with the ether extracts (2 x 30 ml.) of the aqueous layer, was dried

[1960] Decalin and Friedel—Crafts Acetylating Agent. Part II. 4717

(1C2CO8) and the ether was removed under reduced pressure. The residue was washed with cold light petroleum and gave the required hydroxy-ketone (3.4 g.) which crystallised from light petroleum (b. p. 40-60°) in prisms, m. p. 62-63.5° (Found: C, 73.7; H, 10.0. C₁₂H₂₀O₂ requires C, 73.5; H, 10.2%). It showed absorption bands at 1698 (>C=O) and 3460 (OH) cm.⁻¹; the latter was narrow, of high intensity, and independent of concentration, as is required for intramolecular hydrogen bonding. When heated at 90°, the hydroxy-ketone evolved water vapour which was collected on a cold finger and identified. The residual oil was dissolved in light petroleum, dried (CO₂), and passed down an alumina column. Pure vinyl ether (V)

(0.5 g.) was the first fraction eluted and was identified by its infrared absorption spectrum.

Referee #2 (Remarks to the Author):

The authors have further revised their manuscript and explained the selectivity issue. In my opinion, the selectivity is reasonable and the manuscript seems Nature-worthy.

Can the authors include the actual yields for the observed enone byproducts in the manuscript figure, rather than providing a range (similar to what they provided in the experimental document).

I also wondered about their new definition of "charge relocation". The phrase "through a device introduced by synthesis" is unclear and I suggest removing or further editing that phrase.

Author Rebuttals to Second Revision:

Referees' comments:

Referee #1 (Remarks to the Author):

The authors have made several changes to the manuscript to reflect concerns raised during the previous round of peer review. Most significantly, they now address the issue of enone byproduct formation for every example reported in the paper. To summarize their findings, it is not a general phenomenon and seems to become more pronounced with certain acylium cations (in a manner that is not straightforward to rationalize). The percent enone formation is now reported in the footnotes and in fully annotated versions of the scope table that are included in the SI. The presence of these byproducts is now also discussed in the main text. From my perspective, this issue has been thoroughly addressed, and my concerns are allayed.

Other concerns (more minor in nature) still persist and are listed below:

(1) Regarding the term "charge location", again I will defer to the editor on this matter, but I'm not convinced there is a need for inventing this term to describe a phenomenon that has already been described many times using more conventional and established terminology, hydride shift(s). Do the authors view charge relocation as being a more general phenomenon beyond hydride shifts (and possibly alkyl shifts)?

I don't necessarily find issue with creating a term that the authors think will be more appealing to a broad audience, but then when defined upon first use, it should be clear that in this context, "charge relocation" is taking place via successive hydride shifts. (Note that "hydride shift" is never used in the manuscript, despite being the clearest terminology and the terminology established previously for this exact process.)

As mentioned above, and again we thank the editor for his suggestion, this has been addressed.

(2) The claim that the decalin-derived vinyl ether had never been hydrolyzed with a

reported yield until 2011 does not appear to be true.

The same authors as the 1959 paper in question reported follow up studies where yields for this step are also included (see, Baddeley et al., *J. Chem. Soc.* 1960, 4713). Excerpt reproduced below for convenience.

10p-Acetyl-trans-decal-1(3-ol (VI).—A mixture of the vinyl ether (4.5 g.) in ether (30 ml.) and dilute sulphuric acid (60 ml.; x) was stirred and gently refluxed for 2.5 hr. The ether layer was separated and, with the ether extracts (2 x 30 ml.) of the aqueous layer, was dried

[1960] Decalin and Friedel—Crafts Acetylating Agent. Part II. 4717

(1C2CO8) and the ether was removed under reduced pressure. The residue was washed with cold light petroleum and gave the required hydroxy-ketone (3.4 g.) which crystallised from light petroleum (b. p. 40-60°) in prisms, m. p. 62-63.5° (Found: C, 73.7; H, 10.0. C₁₂H₂₀O₂ requires C, 73.5; H, 10.2%). It showed absorption bands at 1698 (>C=O) and 3460 (OH) cm.⁻¹; the latter was narrow, of high intensity, and independent of concentration, as is required for intramolecular hydrogen bonding. When heated at 90°, the hydroxy-ketone evolved water vapour which was collected on a cold finger and identified. The residual oil was dissolved in light petroleum, dried (CO₂), and passed down an alumina column. Pure vinyl ether (V) (0.5 g.) was the first fraction eluted and was identified by its infrared absorption spectrum.

The information within the Figure stated that the initial reports from 1959 (*Proc. Chem. Soc.* 1959, 377-414 and *J. Chem. Soc.* 1959, 1324-1327) had no reported yields for the overall transformation. Indeed, the reviewer's cited reference provides additional information, which has allowed us to calculate the "original" yield for this overall process. Based on the numbers available, we have calculated the yield over 2 steps to be 15% (for reference, the Synlett 2011, which we had originally cited reports 13%). We have now included the yield of 15% and replaced the Synlett 2011 reference with the one pointed out by the reviewer (*J. Chem. Soc.* 1960, 4713-4719).

Referee #2 (Remarks to the Author):

The authors have further revised their manuscript and explained the selectivity issue. In my opinion, the selectivity is reasonable and the manuscript seems Nature-worthy.

Can the authors include the actual yields for the observed enone byproducts in the manuscript figure, rather than providing a range (similar to what they provided in the experimental document).

We have now included the actual yields of the enone by-products, rather than a range. For Figures 2 and 3, these are contained within the captions – seeing as we have convinced the referees that enone formation is not a general problem of our method, we hope that our decision not to highlight the few instances within the figures themselves is accepted by the editor. Moreover, reiterating a point made during previous revisions, we believe that inclusion within the figures themselves might insinuate that, similarly to regioisomers, the by-products are not separable from the desired products, which is not the case (all enones were easily separable by standard FCC).

The figures included in the SI and the Extended Data Fig. 1, however, contain them within.

I also wondered about their new definition of "charge relocation". The phrase "through a device introduced by synthesis" is unclear and I suggest removing or further editing that phrase.

As mentioned above, this has been addressed.